# Digital Addiction in Children and Affecting Factors

**DOI:** 10.3390/children11040417

**Published:** 2024-04-01

**Authors:** Duygu Oktay, Candan Ozturk

**Affiliations:** Department of Child Health and Pediatric Diseases Nursing, Faculty of Nursing, Near East University, TRNC Mersin 10, Nicosia 99138, Turkey; candan.ozturk@neu.edu.tr

**Keywords:** digital addiction, children, scale, student

## Abstract

This study was conducted to identify the digital addiction levels of secondary school students in Northern Cyprus and the factors affecting them. The study was a descriptive, cross-sectional and methodological study. In this study, seven primary and secondary schools were selected randomly in Northern Cyprus and a total of 844 students were included in the study. The children scored 48.12 ± 17.46 points on the “Digital Addiction Scale for Children”. Gender, economic status, age, reasons for using the internet, own cell phone, duration of cell phone and computer use, own computer, mother’s education and place of residence significantly affected the children’s digital addiction level (*p* < 0.05). Digital tools have entered many areas of our lives and while they have positive and helpful aspects, their negative aspects are coming to light day by day. Especially with the COVID-19 pandemic, digital tools have entered our lives more and children have encountered digital tools at an earlier age. It is thought that there is a need to warn about the negative and harmful effects of digital tools on children and to inform families more about this issue. It is recommended not to ignore the negative effects of digital tools and to implement practices to prevent addiction under parental supervision.

## 1. Introduction

With the advancement of technology, easy-to-carry digital tools (phones, tablets, computers, etc.) have brought humanity into the digital age. These digital devices have impacted children’s work, entertainment and social communication and have become a necessity for children’s daily learning and living. Children use digital devices anywhere and anytime; for social media, surfing the internet and playing video games. This process leads to the development of digital addiction. In general, digital addiction includes all kinds of addictive behaviors associated with the use of digital devices such as phones, computers, internet, video games and social media [1]. Due to the fact that children are vulnerable and have not yet completed their brain development, they tend to become addicted more quickly [1]. Another negative effect of digital tool use is that the balance of the reward–punishment system in the brain is disrupted, leading to addiction. This type of addiction is also called “reward failure syndrome”. For this reason, when it becomes difficult for the child to access digital tools or when they are prevented, they may exhibit addiction-specific behaviors [2].

For children with digital addiction, there is a decrease in participation in real-life environments [3] and negative consequences such as procrastination behavior, distraction and decreases in grade achievement due to poor time management [4]. While the focus has been on the positive aspects of digital technology such as creating a more comfortable and better world, today, the negative aspects that affect personal and social life due to the addiction created by this digital technology have gained importance in research [5].

In a meta-analysis study conducted by Cheng and Li (2014) covering 31 countries, it was stated that internet addiction is a mental health problem in many countries; internet use disorder is seen in the age range of 12–41 years and is most common in the Middle East [6]. According to another meta-analysis study by Fam (2018) [7], the global prevalence of internet game addiction in children aged 10–19 years is 4.6%. Hawi et al. (2019) [8] found that 20% of 11,438 students in grades 7–12 spend 5 or more hours a day on social media, 23% play video games almost every day and 30% use different electronic devices. Of these students who participated in the study, 5% had addiction symptoms [8].

Children’s self-awareness and self-regulation skills are underdeveloped compared to adults [9]. When the research on digital addiction in children is examined, it has been found that digital addiction causes problems such as anxiety, sleep disorders, vision problems, obesity, aggression, psychological problems, hyperactivity, attention deficit and suicide in children [9,10,11].

In their review of digital addiction, Ding and Li (2023) discussed studies for the 0–18 age group and stated that digital-based interventions such as web-based and virtual reality are a promising option to reduce the digital addiction levels of adolescents. They also found that intervention programs significantly reduced digital addiction symptoms, depression and anxiety levels, and further studies with prevention programs are needed [12]. Ding et al. (2024) stated in another study that it would be more beneficial for parents to spend more face-to-face time with their children by reducing the use of digital tools outside working hours [13].

The issue of digital addiction in children has become an increasingly important issue recently. However, since there was not enough data, we tried to provide data on this scale, but since there was no data in the literature, its positive and negative aspects could not be discussed. This study is thought to be important as it will fill this gap and contribute to the literature in this respect.

It is a priority initiative to diagnose children’s tendencies regarding the use of digital devices in order to detect the risk of possible digital addiction early and to prevent problems that may develop. There is no study examining the digital addiction status of children in Northern Cyprus. Therefore, this study was conducted to determine the level of digital addiction and the factors affecting it among middle school students aged 9–12 years in Northern Cyprus.

## 2. Materials and Methods

### 2.1. Research Type

The study was comparative and cross-sectional.

### 2.2. Sample of the Study

The study sample included primary school children aged 9–12 with a total of *n* = 13,641. We needed to reach a sample size of 374 people with the sampling formula. There are 6 districts in TRNC. Three of these districts are large and populated districts. There are 50 schools in these 3 large districts. Each school is assigned a sequence number. Then the first number from the table of simple random numbers was determined by lottery. Then, 7 schools were randomly selected from the specified column, respectively. The population of the research consisted of *n* = 1171 students studying in seven primary schools. The research was conducted in 7 primary schools in TRNC and the study was completed with *n* = 844 students. The response rate was 73%.

### 2.3. Data Collection Tools

Socio-demographic data form and “Digital Addiction Scale for Children” (DASC) were used.

#### 2.3.1. Socio-Demographic Data Form

Questions such as mother’s education, father’s education, age, gender, place of residence, purpose of using the internet, owning a phone, owning a computer, and how much time is spent with these devices were investigated.

#### 2.3.2. Digital Addiction Scale for Children

The scale developed by Hawi, Samaha and Griffiths (2019) consists of 25 items and 9 sub-dimensions. The scale is evaluated on a five-point Likert scale (1: Never–5: Always) with a minimum score of 25 and a maximum score of 125. The higher the score, the higher the level of dependency. The reliability coefficient of the original scale is α = 0.936. Sub-dimension reliability coefficients are as follows: preoccupation; α = 0.61, tolerance; α = 0.67, withdrawal; α = 0.87, problems; α = 0.65, conflict; α = 0.59, deception; α = 0.56, displacement; α = 0.62, relapse; α = 0.59 and mood modification; α = 0.68.

The reliability and validity study of the Turkish version of the DASC was conducted by the researchers. For the adaptation study, the scale was first translated into Turkish and then reviewed by experts for language and content. In line with the recommendations of the experts, changes were made to the scale and it was sent back to the Turkish–English translation. The language-validated scale was administered to 315 students studying in 2 primary schools in TRNC.

The Cronbach’s alpha coefficient of the Turkish scale was 0.89. The fit indices of the scale are RMSEA 0.044, GFI 0.907, NFI 0.816, NNFI (TLI) 0.908, CFI 0.919, IFI 0.921. As a result of the factor analysis, it was determined that the Turkish version of the DASC showed a 5-factor structure (problem, relapse, conflict, preoccupation and deception) different from the original scale and consisted of 25 items as in the original scale. 

### 2.4. Data Evaluation

Statistical analysis of the research data was carried out using the Statistical Package for Social Sciences (SPSS) 26.0 software. Cronbach’s alpha test was performed for the reliability of the responses of the children included in the study sample to the Digital Addiction Scale for Children and the alpha coefficient was calculated as 0.880. 

The distribution of children according to their socio-demographic characteristics, internet, mobile phone and computer usage status was determined by frequency analysis. Descriptive statistics of children’s DASC scores were given. As a result of the Kolmogorov–Smirnov test, it was determined that the scale scores did not fit the normal distribution and non-parametric tests were used in the study. The Mann–Whitney U test was used for two groups, the Kruskal–Wallis H test for more than two groups and an independent sample *t* test was used to compare the Digital Addiction Scale for Children scores according to the gender of the children in the study. 

### 2.5. Ethical Aspects of the Study

Verbal and written informed consent was obtained from the children and their families, and permission was obtained from the directorates of the primary schools where the research was conducted and the Near East University Scientific Research Ethics Committee (YDU/2021/98-1468).

## 3. Results

Of the children who participated in the study, 405 (47.99%) were girls and 439 (52.01%) were boys; the majority were 12-year-old children (353—41.82%). One third of the mothers (269—31.87%) and fathers (263—31.16%) had a university education or higher. Half of the families had a good economic status (436—51.66%) and about three quarters lived in the city center (615—72.87%). Most of the children (440—52.13%) used the Internet for doing homework, playing games and doing research, and 613 (72.63%) had their own cell phones. When the duration of time spent by the children with their cell phones was analyzed, it was concluded that the majority 228 (37.19%) spent two hours, 435 (51.54%) of the children had their own computer and 253 (58.16%) spent 1 h on the computer.

The children scored 48.12 ± 17.46 points from the total “Digital Addiction Scale for Children”. When the sub-dimension scores of the Digital Addiction Scale were analyzed, relapse; 15.69 ± 5.69, tolerance; 14.37 ± 6.14, problem; 9.72 ± 4.25, deception; 5.92 ± 2.94 and preoccupation; 6.17 ± 2.84 points were obtained (Table 1).

Statistically significant differences (*p* < 0.05) were found between the scores of the children on the total Digital Addiction Scale and all the sub-dimensions of the scale according to their age (Table 2) and reasons for using the internet (Table 3). It was determined that it had a medium-level effect according to age and a low-level effect according to the reason for internet use.

There were statistically significant differences (*p* < 0.05) between the scores of the children according to economic status (Table 4) and to gender (Table 5) in the total Digital Addiction Scale and in the sub-dimensions of relapse, tolerance, problem and preoccupation. Its impact was low compared to the economic level.

Statistically significant differences (*p* < 0.05) were found between the scores of the children on the total Digital Addiction Scale and all the sub-dimensions of the scale according to their age (Table 2), reasons for using the internet (Table 3), having their own mobile phone and computer and duration of mobile phone and computer use. There were statistically significant differences (*p* < 0.05) between the scores of the children according to gender and economic status (Table 4) in the total Digital Addiction Scale and in the sub-dimensions of relapse, tolerance, problem and preoccupation.

There were statistically significant differences (*p* < 0.05) between the scores obtained from the total Digital Addiction Scale for Children and the relapse, tolerance and problem sub-dimensions of the scale according to the educational status of the mother. Additionally, there were not statistically significant differences (*p* < 0.05) between the scores obtained from the total Digital Addiction Scale for Children in the all sub-dimensions of the scale according to the educational status of the father.

It was determined that there were statistically significant differences between the scores obtained from the total Digital Addiction Scale and the relapse and preoccupation sub-dimensions of the scale according to the place where the children lived (*p* < 0.05) (Table 6).

## 4. Discussion

The descriptive statistics for children’s DASC scores are given in Table 1. The children participating in the study scored 15.69 ± 5.69 points for relapse, 14.37 ± 6.14 points for tolerance, 9.72 ± 4.25 points for problem, 9.72 ± 4.25 points for deception 5.92 ± 2.94 and 6.17 ± 2.84 points for preoccupation. The children scored 48.12 ± 17.46 points on the Digital Addiction Scale for Children (Table 1). The scores of the children from both the scale score and the subscales were below the median. In this sample, the addiction levels of the children were low, and as the score they received from the scale increased, the children’s addiction levels also increased. For this reason, it was determined that the addiction level of the children remained low.

When our study was analyzed based on gender variables, it was determined that the mean scores of the male students’ digital addiction levels were higher than the female students. When digital addiction levels are examined according to the gender variable, studies [14,15,16] have been obtained that male students have higher levels of addiction than female students. Our study was similar to the literature. This similarity was thought to be due to the tolerance shown to boys in societies.

When our study was analyzed based on the age variable, it showed a significant difference in the total scale and all the sub-dimensions. Ten-year-old students scored the highest in the relapse, tolerance and problem sub-dimensions, and twelve-year-old children scored the highest in the deception and preoccupation sub-dimensions. The lowest scores were obtained by eleven-year-old children in the sub-dimensions of relapse, tolerance, problem and deception, and by nine-year-old students in the sub-dimension of preoccupation. Altun and Atasoy (2018) stated in their study that children between the ages of 10–14 were highly dependent [17]. The results of the literature are similar to our study. It is thought that this similarity is due to the fact that children in this age group use the internet more easily. In this study, age was found to have a moderate effect. This result shows that age is an important variable in determining digital addiction. This result reveals that age should be taken into account in diagnoses and interventions for digital addiction.

When the results were analyzed based on the variable of mother’s education, it was concluded that the lowest level was found in the students whose mothers had university and higher education, and the highest level was found in children whose mothers did not go to school. Arslan (2019), in his study on secondary school students, stated that children whose mothers were university graduates had higher levels of digital addiction, while students whose mothers were secondary school graduates had lower levels of digital addiction [14]. Tansel and Bircan Bodur (2021) stated in their study in Turkey that education level and income level are directly proportional [18]. It is thought that the difference between the studies stems from income level, socialization and sociocultural dimensions. 

When analyzed based on the variable of father’s education, the digital addiction levels of children whose fathers had university and higher education were at the lowest level, while those of children whose fathers had never attended school were at the highest level.

Arslan (2019), in a study conducted on middle school students, stated that the father’s education status was effective on digital addiction, and that children whose fathers were university graduates had higher levels of digital addiction, while the group whose fathers were middle school graduates had low levels of digital addiction [14]. While the students with moderate economic status received the highest score, the students with poor economic status received the lowest score. In his study, Arslan concluded that the group with medium economic status had the highest average score and the group with low economic status had the lowest average score according to the total scores [14]. Kayri and Günüc (2016), in their study on the relationship between economic level and internet addiction, stated that students with good economic status have more internet addiction [19]. Our study is similar to the studies of Arslan (2019) [14] and Kayri and Günüc (2016) [19]. It is thought that this similarity is due to the ease of access to digital tools with increasing economic income level. The effect size of the economic level was determined to be low. Although the economic level is low, it still needs to be taken into account in definitions and initiatives regarding digital addiction.

Among the students, those living in the village scored lower on the scale, while those living in the city center scored higher. Celik and Celik (2023), in their metasynthesis study on digital addiction in children, stated that the place of residence is the precursor of digital addiction [20]. Li et al. (2014) concluded in their study that 81.8% of students living in the city use the internet, while only 48.5% of students living in rural areas use the internet [16]. Our study results are similar. It is thought that this similarity is due to the difficulty in accessing digital tools in the village environment.

When the reasons for the children’s use of the internet were examined, the lowest score according to the total score of the scale was to conduct research, and the highest score was to play games, conduct research and do homework. Celik and Celik (2023) stated that the purpose of children’s use of these tools is play and entertainment [20]. Li et al. (2014) stated that the purposes of children’s use of the internet are research, online communication and playing games [16]. Ko et al. (2005) stated that children use the internet to play games, watch movies and listen to music [21]. Biddiss and Irwin (2010) concluded in their study that most of the children are on the internet for the purpose of playing games [22]. Our study is similar to the literature. This similarity is thought to be due to the fact that children take each other as an example. The effect size of the reasons for internet use has a low effect size. It is low but needs to be carefully evaluated for future research.

When examined in terms of the variables of the children’s own cell phone and duration of cell phone use, a significant difference was observed in all the sub-dimensions. The children who had a cell phone and spent three hours or more with their cell phones received the highest score on the scale. The lowest score was obtained by the children who did not have a cell phone, and the lowest score was obtained by the children who spent one hour with their cell phones when analyzed in terms of duration of use. 

According to the results of our study, the duration of use increases in the presence of a cell phone. In the study conducted by Ustundag (2022) on children, it was stated that the addiction levels of children with cell phones were high [23]. Our study results are similar to the literature. It is thought that this similarity is due to the fact that the child spends more uncontrolled time in the presence of his/her own cell phone.

More than half of the students had their own computers. According to the total score of the scale, the children who had their own computers scored high and the children who did not have their own computers scored low. Bagatarhan and Siyes (2023) [24] and Biddiss and Irwin (2010) concluded that the majority of children had a computer [22]. Our study results are similar. This similarity is thought to be due to the similarity of the sample age group. 

The children who spent one hour with their computer received the highest score in the variable of computer usage time, and children who did not have a computer received the lowest score. Li et al. (2014) concluded in their study on primary and secondary school students that most of the children spent less than one hour on the internet. Our study is similar to the literature [16].

This study has limitations arising from its cross-sectional design. Specifically, it provides valuable insights into digital addiction among children in Northern Cyprus, it does not allow for the observation of changes over time or causal relationships.

## 5. Conclusions

Digital tools have entered many areas of our lives and while they have positive and helpful aspects, their negative aspects are coming to light day by day. These results do not mean that technology is completely negative and bad. It is necessary to create early intervention programs to prevent the use of wrong and negative digital tools and to treat digital addiction if it is in question. When the research on digital addiction was examined, it was observed that most of the studies were conducted on adolescents and adults. It is thought that research on digital addiction in children should be increased. Especially with the COVID-19 pandemic, digital tools have entered our lives more and children have encountered digital tools at an earlier age. It is thought that there is a need to warn about the negative and harmful effects of digital tools on children and to inform families more about this issue. It is recommended not to ignore the negative effects of digital tools and to implement practices to prevent addiction under parental supervision.

Limitation: No study using the DASC scale was found. Since no study on this subject could be found in the literature, the results could not be compared with existing research. However, considering that this is the first study conducted on a large sample and yielded positive results, it is believed that the impact of these limitations has been mitigated.

## Figures and Tables

**Table 1 children-11-00417-t001:** DASC scores.

	*n*	$\bar{x}$	SD	M	IQR	Min	Max
Relapse	844	15.69	5.69	15	8	7	35
Tolerance	844	14.37	6.14	9	6	7	35
Problem	844	9.72	4.25	9	6	5	25
Deception	844	5.92	2.94	5	4	3	15
Preoccupation	844	6.17	2.84	6	3	3	15
DASC total score	844	51.87	18.25	44	24	25	125

**Table 2 children-11-00417-t002:** Comparison of DASC scores by age.

	Age	*n*	$\bar{x}$	SD	Median	Mean Rank	X^2^	*p*	The Difference (r)
Relapse	9	166	15.07	5.88	14	395.53	10,839	0.013 *	2–3 (0.153)
	10	144	17.13	7.64	16	462.76			
	11	181	14.92	5.54	14	387.78			
	12	353	15.93	5.27	15	436.56			
Tolerance	9	166	11.05	5.07	9	450.34	15,004	0.002 *	1–3 (0.178)
	10	144	11.15	5.17	9.5	452.20			2–3 (0.180)
	11	181	9.37	4.35	9	363.97			3–4 (0.123)
	12	353	10.72	6.23	9	427.30			
Problem	9	166	9.60	4.66	9	400.67	25,326	0.000 *	2–3 (0.210)
	10	144	10.42	4.54	9	459.56			3–4 (0.189)
	11	181	8.56	3.71	8	352.91			
	12	353	10.07	4.08	9	453.33			
Deception	9	166	5.56	2.87	5	387.69	39,643	0.000 *	2–3 (0.196)
	10	144	6.25	3.31	5	438.98			3–4 (0.256)
	11	181	5.03	2.52	4	342.99			1–4 (0.166)
	12	353	6.40	2.89	6	472.92			
Preoccupation	9	166	5.75	2.80	5	380.96	15,679	0.001 *	1–4 (0.166)
	10	144	6.46	3.44	6	425.17			3–4 (0.132)
	11	181	5.77	2.64	5	390.13			
	12	353	6.46	2.65	6	457.54			
DASC total score	9	166	47.04	16.95	42	406.47	20,042	0.000 *	2–3 (0.205)
	10	144	51.41	19.61	49	459.63			3–4 (0.171)
	11	181	43.66	15.31	40	359.01			
	12	353	49.58	17.38	45	447.45			

* *p* < 0.05 (Kruskal–Wallis H test).

**Table 3 children-11-00417-t003:** Comparison of DASC scores based on the reasons for children’s internet use.

	Reasons	*n*	$\bar{x}$	SD	Median	Mean Rank	X^2^	*p*	The Difference (d **)
Relapse	Doing homework	113	14.35	5.15	14	370.34	66,599	0.000 *	1–2 (0.378)
	Playing games	142	18.96	6.15	19	555.58			2–3 (0.452)
	Doing research	149	14.07	6.80	13	335.68			2–4 (0.235)
	All of them	440	15.64	5.35	15	422.35			
Tolerance	Doing homework	113	10.12	4.63	9	403.45	28,884	0.000 *	1–2 (0.293)
	Playing games	142	12.55	5.43	13	517.60			2–3 (0.233)
	Doing research	149	9.78	6.01	9	375.11			2–4 (0.185)
	All of them	440	10.31	5.42	9	412.75			
Problem	Doing homework	113	8.96	3.88	8	378.12	41,716	0.000 *	1–2 (0.327)
	Playing games	142	11.92	4.83	12	538.13			2–3 (0.328)
	Doing research	149	9.03	4.05	8	379.61			2–4 (0.225)
	All of them	440	9.43	3.98	9	411.10			
Deception	Doing homework	113	5.76	2.79	5	412.31	47,831	0.000 *	1–2 (0.280)
	Playing games	142	7.61	3.40	7	547.76			2–3 (0.351)
	Doing research	149	5.40	2.67	4	379.51			2–4 (0.266)
	All of them	440	5.59	2.71	5	399.25			
Preoccupation	Doing homework	113	5.47	2.80	4	352.25	51,582	0.000 *	1–2 (0.374)
	Playing games	142	7.66	3.36	7	533.68			2–3 (0.370)
	Doing research	149	5.44	2.61	5	355.33			2–4 (0.190)
	All of them	440	6.12	2.56	6	427.41			
DASC total score	Doing homework	113	44.65	15.38	41	375.31	65,894	0.000 *	1–2 (0.383)
	Playing games	142	58.70	19.07	58.5	563.10			2–3 (0.442)
	Doing research	149	43.72	17.64	39	347.97			2–4 (0.262)
	All of them	440	47.09	16.02	44	414.48			

* *p* < 0.05 (Kruskal–Wallis H test), ** Cohen (d).

**Table 4 children-11-00417-t004:** Comparison of DASC scores by the economic status of children.

	Economic Status	*n*	$\bar{x}$	SD	Median	Mean Rank	X^2^	*p*	The Difference (r)
Relapse	Poor	47	14.81	6.02	13	375.13	14,570	0.001 *	1–2 (0.111)
	Middle	361	16.40	5.43	16	459.04			
	Good	436	15.31	6.30	14	397.35			
Tolerance	Poor	47	9.43	4.57	8	362.59	6130	0.047 *	1–2 (0.105)
	Middle	361	10.82	5.52	10	442.66			
	Good	436	10.48	5.56	9	412.26			
Problem	Poor	47	8.87	4.65	8	351.28	12,172	0.002 *	1–2 (0.134)
	Middle	361	10.08	4.03	9	453.19			
	Good	436	9.50	4.36	9	404.76			
Deception	Poor	47	5.02	2.38	4	349.63	4970	0.083	
	Middle	361	6.04	2.98	5	432.54			
	Good	436	5.92	2.94	5	422.04			
Preoccupation	Poor	47	5.70	2.64	5	382.28	7082	0.029 *	1–2 (0.096)
	Middle	361	6.45	2.91	6	447.26			
	Good	436	5.99	2.79	6	406.33			
DASC total score	Poor	47	43.83	17.70	40	351.31	12,630	0.002 *	1–2 (0.135)
	Middle	361	49.80	16.75	47	454.17			
	Good	436	47.20	17.90	43	403.95			

* *p* < 0.05 (Kruskal–Wallis H test).

**Table 5 children-11-00417-t005:** Comparison of Digital Addiction Scale for Children Scale scores by gender of children.

	Gender	*n*	$\bar{x}$	SD	Median	Mean Rank	Z	*p*
Relapse	Female	405	14.73	5.37	14	380.73	−4.789	0.000 *
	Male	439	16.69	6.30	16	461.04		
Tolerance	Female	405	10.01	5.24	9	393.88	−3.291	0.001 *
	Male	439	11.08	5.69	10	448.91		
Problem	Female	405	9.04	3.88	8	385.37	−4.273	0.000 *
	Male	439	10.34	4.48	9	456.76		
Deception	Female	405	5.72	2.80	5	408.22	−1.661	0.097
	Male	439	6.10	3.05	5	435.67		
Preoccupation	Female	405	5.79	2.68	5	388.76	−3.907	0.000 *
	Male	439	6.53	2.94	6	453.63		
Digital Addiction Scale For Children	Female	405	45.29	16.30	42	382.23	−4.610	0.000 *
	Male	439	50.74	18.10	47	459.65		

* *p* < 0.05.

**Table 6 children-11-00417-t006:** Comparison of children’s Digital Addiction Scale scores by place of residence.

	Residence	*n*	$\bar{x}$	SD	Median	Mean Rank	Z	*p*
Relapse	Village	229	14.68	5.85	14	374.04	−3.530	0.000 *
	Town	615	16.15	5.94	15	440.54		
Tolerance	Village	229	10.28	5.91	9	408.98	−0.988	0.323
	Town	615	10.68	5.34	9	427.53		
Problem	Village	229	9.55	4.16	9	412.87	−0.704	0.481
	Town	615	9.78	4.28	9	426.09		
Deception	Village	229	5.84	2.96	5	412.54	−0.736	0.462
	Town	615	5.95	2.93	5	426.21		
Preoccupation	Village	229	5.83	2.82	5	389.03	−2.462	0.014 *
	Town	615	6.30	2.84	6	434.96		
Digital Addiction Scale for Children	Village	229	46.17	17.70	41	390.45	−2.331	0.020 *
	Town	615	48.85	17.33	45	434.43		

* *p* < 0.05 (Mann–Whitney U test).

## Data Availability

Data is contained within the article.

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
