# Peer review of "Digital Addiction in Children and Affecting Factors"

_children, 2024, doi:10.3390/children11040417_

Round 1
Reviewer 1 Report
Comments and Suggestions for Authors
This paper analyzes the level of digital addiction among children aged 9-12.
The introduction is focused only on the general data with a general description of the phenomenon. There should be mentions of other studies, especially those which have used the DASC. Although authors describe the study as descriptive, there is a clear hypothesis testing using non parametric statistics. The rationale for forming the hypotheses must be clearly stated to prevent “fishing” of the p-values.
The paper lacks the procedure chapter. How were the schools selected and what was the procedure of collecting the data.
The sample in question is a large sample where due to the sensitivity of KS test it should not be expected to have normal distributions. However the results would benefit from analysing other indicators of normality, such as skewness and kurtosis to determine the appropriateness of parametric statistics (e.g. if the distributions are not normal then table 1 should have median and quartiles).
Authors here show four variables for testing of differences: age, reasons for internet use, economic status, and urbanicity. In line 157 authors state they have tested the differences according to gender but there are no results shown.
In table 2 there is no description of the question formulation. The authors state that this is the reason for children’s internet use, and from the context I presume that the participants had to choose one of the several options. There should be an explanation why the authors opted for this type od question instead of the quantitative evaluation of each choice which would be more informative (e.g. Doing homework 1-5). Please provide effect sizes along with the results and make the interpretations in line with the effect sizes.
There is no discussion regarding the total score on the DASC, is this a high or low score, how does it compare to other relevant studies.
Author Response
Dear Reviewer,
Thank you for your thoughtful and detailed feedback on our article ( ID children-2898017) titled "Digital Addiction in Children and Affecting Factors" We wish to express our appreciation to the Reviewer for their insightful comments, which have helped us significantly to improve our manuscript. According to the suggestions, we have thoroughly revised our manuscript and its final version is enclosed. Point-by-point responses to the comments listed below.
Yours sincerely
Duygu Oktay

Reviewer 2 Report
Comments and Suggestions for Authors
Review:
DIGITAL ADDICTION IN CHILDREN AND AFFECTING 2 FACTORS
Content:
This paper identifies the digital addiction levels of secondary school students in Northern Cyprus and the factors affecting them.
plus:
The study has a clear structure, including methods, results, tables. discussion and conclusions.
Very current and important topic.
Important statements and results:
line 36-37: "Another negative effect of digital tool use is that the balance of the reward-punishment system in the brain is disrupted, leading to addiction. This type of addiction is also called "reward failure syndrome".
222 : Among the students, those living in the village scored lower on the scale, while those living in the city center scored higher.
268-269 Especially with the Covid-19 pandemic, digital tools have entered our lives more and children have encountered digital tools at an earlier age
challenge for readers:
line 264-265 : "It is necessary to create early intervention programs to prevent the use of wrong and negative digital tools and to treat digital addiction if it is in question."
It would be good to implement 2 studies related to Digital addictions, COvid and challenges.
Tkácová, H.; Pavlíková, M.; Stranovská, E.; Králik, R. Individual (Non) Resilience of University Students to Digital Media Manipulation after COVID-19 (Case Study of Slovak Initiatives). Int. J. Environ. Res. Public Health 2023, 20, 1605. https://doi.org/10.3390/ijerph20021605
Sirotkin, A.V.; Pavlíková, M.; Hlad, Ľ.; Králik, R.; Zarnadze, I.; Zarnadze, S.; Petrikovičová, L. Impact of COVID-19 on University Activities: Comparison of Experiences from Slovakia and Georgia. Sustainability 2023, 15, 1897. https://doi.org/10.3390/su15031897
and very important is FAMILY and FREE TIME:
Kralik, R. (2023). The Influence of Family and School in Shaping the Values of Children and Young People in the Theory of Free Time and Pedagogy. Journal of Education Culture and Society, 14(1), 249-268. https://doi.org/10.15503/jecs2023.1.249.268
please check - again :
?
line: 185
to the literature. - to literature
line 181
gender variable, - gender variables
authors: congratulation to your article.
Author Response

(The authors gave the same response as above.)

Reviewer 3 Report
Comments and Suggestions for Authors
The study aims to identify the levels of digital addiction among secondary school students in Northern Cyprus and the factors influencing it. This question is both relevant and timely, given the increasing integration of digital tools in children's lives. Compared to existing literature, this study adds new insights by focusing on a relatively understudied population and region. It provides empirical data on digital addiction levels and contributing factors, which could inform interventions and policy.
As a reviewer, I suggest that the Authors include a section in their discussion acknowledging the limitations inherent to their study's cross-sectional design. Specifically, it would be beneficial for the authors to note that while their methodology provides valuable insights into digital addiction among children in Northern Cyprus, it does not allow for the observation of changes over time or causal relationships. Highlighting these limitations not only demonstrates rigor and transparency but also helps contextualize the findings within the broader research landscape. Moreover, suggesting potential directions for future research, such as employing a longitudinal design, could further contribute to the field's understanding of digital addiction's dynamics.
Author Response

(The authors gave the same response as above.)

Round 2
Reviewer 1 Report
Comments and Suggestions for Authors
The authors made the edicts that filfill the expectrations enough
Reviewer 3 Report
Comments and Suggestions for Authors
I would like to thank the Authors for all the improvements. I have no further comments.